# Gene Therapy Using Efficient Direct Lineage Reprogramming Technology for Neurological Diseases

**DOI:** 10.3390/nano13101680

**Published:** 2023-05-19

**Authors:** Yujung Chang, Sungwoo Lee, Jieun Kim, Chunggoo Kim, Hyun Soo Shim, Seung Eun Lee, Hyeok Ju Park, Jeongwon Kim, Soohyun Lee, Yong Kyu Lee, Sungho Park, Junsang Yoo

**Affiliations:** 1Laboratory of Regenerative Medicine for Neurodegenerative Disease, Stand Up Therapeutics, Hannamdaero 98, Seoul 04418, Republic of Korea; drchang@stutps.com (Y.C.);; 2Department of Molecular Biology, Nuturn Science, Sinsadong 559-8, Seoul 06037, Republic of Korea; 3Department of Chemistry, Sungkyunkwan University, 2066, Seobu-ro, Jangan-gu, Suwon-si 16419, Republic of Korea; sungwoo22@skku.edu (S.L.); kjw1029@skku.edu (J.K.); s11hy1n@skku.edu (S.L.); 4Department of Bio-Health Technology, College of Biomedical Science, Kangwon National University, 1 Kangwondeahak-gil, Chuncheon 24341, Republic of Korea; jieunkim@kangwon.ac.kr; 5Research Animal Resource Center, Korea Institute of Science and Technology, Hwarang-ro 14-gil, Seongbuk-gu, Seoul 02792, Republic of Korea; selee@kist.re.kr; 6Database Laboratory, Department of Computer Science and Engineering, Dongguk University-Seoul, Pildong-ro 1-gil 30, Jung-gu, Seoul 04620, Republic of Korea; phj1987@gmail.com (H.J.P.); yklee@dongguk.edu (Y.K.L.)

**Keywords:** cell fate conversion, direct lineage reprogramming, spinal cord injury, gene therapy, nanoporous particle-based gene delivery

## Abstract

Gene therapy is an innovative approach in the field of regenerative medicine. This therapy entails the transfer of genetic material into a patient’s cells to treat diseases. In particular, gene therapy for neurological diseases has recently achieved significant progress, with numerous studies investigating the use of adeno-associated viruses for the targeted delivery of therapeutic genetic fragments. This approach has potential applications for treating incurable diseases, including paralysis and motor impairment caused by spinal cord injury and Parkinson’s disease, and it is characterized by dopaminergic neuron degeneration. Recently, several studies have explored the potential of direct lineage reprogramming (DLR) for treating incurable diseases, and highlighted the advantages of DLR over conventional stem cell therapy. However, application of DLR technology in clinical practice is hindered by its low efficiency compared with cell therapy using stem cell differentiation. To overcome this limitation, researchers have explored various strategies such as the efficiency of DLR. In this study, we focused on innovative strategies, including the use of a nanoporous particle-based gene delivery system to improve the reprogramming efficiency of DLR-induced neurons. We believe that discussing these approaches can facilitate the development of more effective gene therapies for neurological disorders.

## 1. Introduction

Cell fate conversion technology originated with the generation of induced pluripotent stem cells (iPSCs) via viral transduction of specific transcription factors, namely, octamer-binding transcription factor 4, SRY-Box transcription factor 2, KLF transcription factor 4, and MYC proto-oncogene, bHLH transcription factor (OSKM) [1]. This technology demonstrated that a fully differentiated somatic cell could be reprogrammed into a completely different cell type by introducing key transcriptional factors of starting cells. Based on this principle, numerous studies have shown the feasibility of direct lineage reprogramming (DLR) from fibroblasts to various cell types, such as neurons, cardiomyocytes, chondrocytes, and hepatocytes.

Specifically, for neuronal reprogramming, DLR technology can be used to reprogram fibroblasts into various neuronal types, including cholinergic motor, glutamatergic, GABAergic, and dopaminergic neurons [2,3,4,5,6]. In 2010, it was revealed that the ectopic expressions of achaete-scute homolog 1 (ASCL1), POU domain transcription factor (BRN2), and myelin transcription factor 1-like (MYT1L) (ABM) could induce DLR of fibroblasts into functional glutamatergic neurons, characterized by synaptic activity and the regulation of Na^+^/K^+^ currents. In 2014, Wernig et al. showed that ASCL1 alone could induce DLR of functional glutamatergic neurons, which demonstrated that ASCL1 is the key master regulator that drives neuronal lineage development, with the assistance of two other factors.

Another example is the generation of induced dopaminergic neurons (iDNs). To generate iDNs, it is crucial to cotransduce dopaminergic lineage transcriptional factors, nuclear receptor 4A2 (NURR1), paired-like homeodomain 3 (PITX3), and LIM homeobox transcription factor 1 alpha (LMX1A) with ASCL1 (ANPL) [7]. By generating iDNs using DLR technology, researchers have provided evidence that Parkinson’s disease (PD) could be treated. PD is a progressive neurodegenerative disease that mainly affects motor function. It is characterized by the degeneration of dopaminergic neurons in the substantia nigra-to-striatum circuit, which leads to neurotransmitter imbalances and Parkinsonian symptoms. The cardinal symptoms of this disease include bradykinesia (slowness of movement), rigidity (stiffness and increased resistance to passive limb movements), tremor (involuntary and rhythmic shaking), and postural instability (impaired balance and coordination). In research settings, 1-methyl-4-phenyl-1,2,3,6-tetrahydropyridine (MPTP) and 6-hydroxydopamine hydrobromide (6-OHDA)-induced PD mouse models exhibited a lack of dopaminergic neurons and generated small amounts of dopamine, thus, displaying a Parkinsonian phenotype. The conversion of astrocytes into dopaminergic neurons in the striatum region led to the production of tyrosine hydroxylase (TH)-positive cells from originally glial fibrillary acidic protein (GFAP)-positive cells, which reduced Parkinsonian symptoms.

Spinal cord injury (SCI) is another major neurological disorder that can cause movement disabilities. SCI is a devastating condition that can lead to partial or complete loss of motor and sensory functions below the level of an injury. The pathophysiology of the disease involves complex interplay among secondary injury mechanisms, including oxidative stress, inflammation, cellular toxicity, and demyelination. Despite extensive research efforts, there is currently no effective therapy for SCI. Gene therapy is a promising approach as it allows targeted delivery of therapeutic genes to an injured spinal cord.

Motor neurons are specialized neurons that control the contraction of skeletal muscles, which permits movement and locomotion. In SCI, the motor neurons that innervate the muscles below the level of injury are often damaged or destroyed, which leads to paralysis and motor malfunction. Recently, the cotransduction of transcriptional factors, ASCL1, BRN2, MYT1L, and LIM homeobox 3 (Lhx3) has been reported to induce DLR of motor neurons from fibroblasts [8].

DLR exhibits several advantages over iPSC redifferentiation techniques because it does not pass through a pluripotent stage, which enables the production of stem cell-free cell sources (e.g., astrocytes and fibroblasts) [5]. Furthermore, DLR can address safety issues, such as teratoma formation, and it is more cost-effective for clinical applications. The OSKM-induced iPSCs possess a risk of differentiating into other types of pluripotent stem cells because the early phase of intermediate iPSCs is unstable. Moreover, the regeneration of iPSCs is less cost effective than DLR because multiple steps are involved in creating iPSCs; thus, dealing with iPSC reprogramming, differentiation, and transplantation to the target organs is difficult. However, in vivo DLR could be more successful for direct differentiation at precise locations in several organs. In vivo DLR is accomplished by manipulating the expression of certain genes in the targeted cells, leading to their conversion into a different cell type without the need for cell transplantation or external manipulation (Figure 1a). In vivo DLR could be used to treat a wide range of diseases by converting the affected cells into healthy cells [9,10]. Guo et al. showed that targeted expression of NeuroD1 to astrocyte induced glutamatergic and GABAergic neurons after brain injury or in Alzheimer’s disease mouse brain [11]. Similarly, NeuroD1-induced reprogramming of astrocytes has been applied for the treatment of a stroke model [12]. In the adult mouse striatum, Cervo et al. showed that combinatorial expression of NeuroD1, Ascl1, Lmx1a, and miR-218 induced dopamine neurons in PD mouse model [13]. These studies provide evidence that in vivo DLR may hold potential for treating neurodegenerative disease by replacing lost or damaged cells with newly converted cells (Figure 1b).

Consequently, more than 100 studies since 2010 have investigated the feasibility of DLR technology for gene therapy. Nonetheless, the low efficiency and quality of reprogrammed cells remain to be significant obstacles to DLR-based gene therapy [14,15,16,17]. Therefore, various studies have attempted to enhance DLR efficiency by treating additional genetic factors, miRNAs, small molecules, or biocompatible materials [5,18,19,20,21,22,23].

The use of nanoparticles as a delivery system in the biomedical field is another promising subject in drug delivery. Improving the stability, bioavailability, and pharmacokinetics of therapeutic agents is possible by encapsulating them within nanoparticles. Moreover, side effects, such as cytotoxicity, inflammation, and reactive oxygen species (ROS) generation, can be reduced. Various types of nanoparticles, including liposomes, polymeric organic nanoparticles, and inorganic nanoparticles, have been employed for targeted drug delivery in cancer therapy. This method enables selective delivery of anticancer drugs to tumor cells while minimizing damage to other tissues [24]. Furthermore, nanoparticles can also be used for the delivery of cDNA, siRNA, and proteins. The use of nanoparticles in a delivery system instead of a viral transduction system or combination of small molecules offers several advantages, including enhanced safety, reduced immunogenicity, tunable physicochemical properties, and the ability to encapsulate and protect genetic material, such as DNA, RNA, and oligonucleotides. Additionally, nanoparticles facilitate targeted delivery and controlled release of therapeutic genes, ultimately improving the therapeutic efficacy and reducing off-target effects [25,26,27].

Nanoparticles can be employed in tissue engineering to construct biomimetic matrices. They are called nanotopographical structure and it can mimic the native extracellular matrix, thereby, providing a conducive environment for cell adhesion, proliferation, and reprogramming [5,28]. For example, a recent study by Yoo et al. demonstrated that an artificially engineered nanotopographical matrix could promote the direct conversion of dopaminergic neurons by modulating the focal adhesion kinase (FAK) molecule in the cell reprogramming process. This modulation increased H3K4me3 levels in the transcription start site of dopaminergic neuronal marker genes, such as tyrosine hydroxylase (Th), dopamine transporter (Dat), and microtubule-associated protein 2 (Map2) [23]. Furthermore, nanoparticles can also serve as catalytic activators for cellular processes, which facilitates cell differentiation or reprogramming. For instance, hydroxyapatite nanoparticles have been employed for the development of bone graft substitutes that promote osteointegration and bone regeneration. Additionally, solid AuNPs with electromagnetic properties have been shown to stimulate the cell reprogramming process by regulating histone acetylation H4K12ac [5].

Reducing oxidative stress is critical for the DLR process because the conversion of cell fate requires significant energy undertaking. A recent study by Lee et al. showed that the use of gold nanoporous rods (AuNpRs) to alleviate cell stress had a considerable impact on the efficacy of induced neuron direct conversion. The researchers observed that AuNpRs functioned as ROS scavengers in DLR, thus, lowering internal cellular conversion stress. Their discovery demonstrated that the use of AuNpRs enhanced the efficiency of DLR-induced dopaminergic (iDA) neurons originating from brain astrocytes, thereby, ameliorating the Parkinsonian phenotype in animal models [29].

Neurological disorders are one of the major interests in the field of gene therapy, with many researchers suggesting the possibility of treating neurological diseases, such as PD and SCI, using DLR technology. The use of master regulators that positively control cellular stress or activate the neuronal actuator (related genes) by opening the related chromatin can serve as a novel therapeutic approach for treating neurological diseases using DLR technology [29,30]. Another way to accelerate DLR is to apply modified nanoparticles. The use of nanotechnology in the biomedical engineering field is emerging as a novel therapeutic approach for treating neurological diseases, with over 10 studies having described how some nanosized gold or graphene nanostructures can facilitate the DLR process [5,14,23,31,32,33,34,35,36].

Nanotechnology has brought significant advancements in the biomedical engineering field and has revolutionized the treatment of various diseases. Various nanotechnology approaches have been implemented in this field including nanocarriers, which have provided numerous experimental and clinical benefits. One of the experimental advantages is their ability to facilitate targeted drug delivery. Nanocarriers such as gold nanoparticles and nanorods can transport growth factors or drugs to specific regions in the brain [37,38,39]. Delivery of targeted growth factors or drugs allows for higher doses of drugs to reach a target area while minimizing toxicity to healthy neurons, improving cargo efficacy, and reducing side effects. In addition, nanotechnology methods have also facilitated the development of regenerative medicine, which aims to repair or replace damaged tissues and organs. Scaffold nanomaterials can mimic the extracellular matrix and provide neuronal structural support for neuronal growth, allowing for the regeneration of neuronal networks in the brain [40,41]. Taken together, nanotechnology methods have significant experimental and clinical importance in the biomedical engineering field.

## 2. Generation of Induced Neurons via Cell Fate Conversion

### 2.1. Acceleration of the Direct Neuronal Reprogramming Process

Cell fate conversion techniques hold substantial promise within the realm of gene therapy for neurological diseases, and therefore, have garnered immense interest among researchers in the field. Consequently, numerous approaches aimed at expediting the process of cell fate conversion have been investigated [5,31,42,43]. Of particular note, cell DLR represents a cutting-edge technology within the field of regenerative medicine. A pioneering study discovered that the direct conversion of induced neurons could be achieved by transducing key transcriptional factors. The ectopic expression of ASCL1, BRN2, and MYT1L (ABM) induced cortical neuronal reprogramming; ASCL1, NURR1, PITX3, and LMX1A (ANPL) induced dopaminergic neuronal reprogramming; and ASCL1, BRN2, MYT1L, and Lhx3 produced motor neurons’ action potential, forming the neuromuscular junction [2,3,44]. Several studies have described various methodologies for promoting neuronal DLR. Table 1 presents an overview of the different types of DLR enhancers, including small molecules, such as CHIR99021, a GSK3 inhibitor; LDN193189, a BMP type 1 receptor ALK2/3 inhibitor; A83-01, a TGF-β type 1 receptor ALK4/5/7 inhibitor; and ascorbic acid. RNA interference (RNAi) using miR-124 and miR-9/9 and biocompatible materials such as AuNpRs, nanotopographical substrates, and graphene nanosheets, have also been shown to improve the efficiency of DLR [5,19,23,31,45,46]. Recent evidence suggests that inorganic materials can regulate cellular mechanisms by modulating epigenetic modifications. Graphene, a single-atom-thick sheet of carbon atoms with a two-dimensional honeycomb structure, possesses unique physical, chemical, and mechanical properties. This materials has been shown to significantly activate H3K4me3 by inducing mesenchymal-to-epithelial transition [28]. Additionally, nanogrooved substrates have been reported to upregulate H3K4me3 and decrease H3K9me3 and H3K27me3 levels, thereby, activating FAK and promoting dopaminergic neuronal fate [23]. Furthermore, the in vivo injection of electromagnetized AuNPs has been documented to significantly activate H4K12 acetylation in histones, which resulted in the generation of iDNs in a mouse model of PD [5].

The use of an appropriate DNA carrier for reprogramming transcriptional factors is necessary for applying DLR technology in the field of gene therapy. Currently, there is significant interest among researchers in using viral vectors for gene therapy, owing to their ability to effectively deliver the necessary genetic material to the target cells. Globally, lentivirus, retrovirus, adenovirus, and adeno-associated virus (AAV) are commonly used viral vectors for ex vivo cell therapy and in vivo gene therapy [47,48,49,50]. There are still some concerns surrounding the use of AAV vectors. One of the concerns is an AAV-induced autoimmune response, which can decrease the effectiveness of the treatment or cause adverse reactions in AAV-treated patients [51]. Additionally, AAV vectors have a limitation since they carry cargoes that are dependent on the size or complexity of the gene [52].

Nonetheless, AAV is a promising candidate for in vivo gene therapy because of its unique features and key advantages over other viral vectors. AAV is known for its nongenomic integration and low toxicity, which significantly alleviate the risk of immunogenicity, mutagenesis, and genotoxicity associated with other viral vectors. For example, systemically administered recombinant AAV used for cancer therapy has shown that rAAVV2/5 has no genotoxicity in nonhuman primate and human livers [53]. In research on the AAV5-cohPBGD vector used in cancer therapy, its intravenous administration had no side effects or integration of immunogenicity and genotoxicity in primate tissues, such as the liver, spleen, and adrenal gland [54]. As a result, AAV has gained widespread attention and is being increasingly used in clinical studies [55,56]. A review of gene therapy trials indicated that 149 clinical trials employed pAAV plasmid vector, of which 94 have been concluded, with 51 reaching the endpoint of efficacy [57]. Most ongoing gene therapy trials are phase 1/2 safety tests. Furthermore, clinical trials that employed AAV serotype 2 were primarily conducted in the field of neurological disorders [58]. AAV2 has the greatest safety and efficacy record in neurological research, as evidenced by over 40 completed clinical trials. The ongoing and completed clinical trials based on AAV2 are listed in Table 2 [58,59,60,61,62,63,64,65,66,67,68,69].

### 2.2. The Role of AuNpRs in the Direct Neuronal Reprogramming Process

The other potential clinical method for gene therapy in the future is to apply biocompatible nanomaterials, particularly AuNpRs. Several studies have shown that specifically modified or functionalized nanoparticles exert crucial effects on biological processes, such as cell survival and differentiation. AuNPs exhibit various therapeutic characteristics, including biocompatibility, ROS scavenging effect, high surface reactivity, and plasmon resonance [29,70,71,72,73]. Not all AuNPs have these effects; however, specifically modified AuNPs or AuNPs integrated with other materials have a drastic impact on the DLR process. Recently, Yoo and Park et al. reported that AuNpRs played a key role as ROS scavengers, ameliorating Parkinsonian phenotypes (i.e., slowness, muscle rigidity, and loss of movement) [29]. They injected AuNpRs into the substantia nigra region in 6-OHDA-induced Parkinsonian mouse model with ASCL1, NURR1, PITX3, and LMX1A, which are dopaminergic neuron conversion transcriptional factors [29]. In the group treated with AuNpRs, a higher number of TH-, DAT-, Forkhead Box A2 (FOXA2)-, and G protein-activated inward rectifier potassium channel 2 (GIRK2)-positive neurons were observed using immunostaining (Figure 2a). To investigate the underlying mechanism, the researchers performed RNA transcriptomic studies during the AuNpR treatment in the reprogramming process. The results revealed an increase in antioxidation-related molecules, particularly ROS scavenging molecules (i.e., Gsta4, Mt3, Sod1, Sod2, and Sirt3), in the AuNpR-treated groups (Figure 2b). Most gene ontology analyses were classified under the oxidation–reduction processes and detoxification.

DA neurons are particularly susceptible to oxidative stress owing to several factors, which contribute to their vulnerability in PD. First, excessive ROS can negatively affect the enzymes involved in dopamine synthesis. TH is the rate-limiting enzyme in dopamine biosynthesis, which converts tyrosine to levodopa. ROS can inhibit TH activity by causing oxidative modification of the enzyme, such as the formation of disulfide bonds and carbonyl groups. This modification results in reduced dopamine production. Second, dopaminergic neurons store dopamine in vesicles via the action of vesicular monoamine transporter (VMAT2). Excessive ROS can impair VMAT2 function, which decreases the vesicular storage of dopamine. This process increases cytosolic dopamine levels, which, in turn, makes more dopamine available for auto-oxidation and subsequent ROS generation, thus, leading to a vicious cycle of oxidative stress and damage. Lastly, dopaminergic neurons themselves are particularly susceptible to oxidative stress because of their high metabolic rate, low antioxidant defenses, and inherent vulnerability associated with dopamine metabolism.

Owing to these reasons, regulating ROS levels during iDA reprogramming is crucial. Yoo and Park et al. successfully demonstrated that antioxidants and ROS scavenging molecules, mediated by AuNpRs, could enhance iDA conversion and alleviate PD phenotypes in animal models [29].

In a study, AuNpRs were synthesized by codepositing Au and Au plating solution at a voltage of −0.95 V to produce Au/Ag alloy nanomaterials. Subsequently, 30% nitric acid was used to dealloy them. To create nanopores in Au nanorods, precise control of the atomic composition of the Au/Ag alloy was achieved by varying the volume ratio of the Au plating solution mixture. To confirm whether AuNpRs were well located in the cell membrane, Yoo and Park attempted to modify the surface of AuNpRs. The surface was modified by incorporating the fluorescence dye rhodamine B (Rho B). They observed that the induced neuron-converted cells exhibited red fluorescence, which could be attributed to the presence of the Rho B dye on AuNpRs. This result showed that the cellular component and biological process could be manipulated by administrating specific types of nanoparticles, such as AuNpRs. Hence, further investigations are being conducted to explore the potential of various shapes and types of nanoparticles in the field of gene therapy and regenerative medicine. Previously, Lee et al. revealed that nanoparticles could be modified by galvanic replacement or the Kirkendall effect [74,75,76]. For example, gold porous lens, gold porous ring, gold shell ring, or gold nanolens have been generated via these reactions (Figure 3) [74,75,76]. In Appendix A, a detailed description of the method for producing AuNpRs (Appendix A) and nanotubes using a three-electrode system is provided. By employing AuNpRs, they successfully generated induced neurons.

### 2.3. The Role of Biocompatible Materials as Delivery Cargo for DLR

Recently, several types of biocompatible materials have emerged as promising DNA carriers for gene therapy [77]. Nanoparticles, such as liposomes, polymeric nanoparticles, and inorganic nanoparticles, have been engineered to improve the efficiency, specificity, and safety of gene delivery to target cells [78,79,80]. Versatile carriers can protect the encapsulated DNA from degradation, enhance cellular uptake, and facilitate controlled release, and thereby, increase the possibility of successful gene delivery [79]. In the context of gene therapy for neurological diseases, the use of nanoparticles as DNA carriers can overcome the challenges associated with crossing the blood–brain barrier, thus, enabling more effective and targeted delivery of therapeutic genes [81]. By leveraging the advances in nanoparticle technology, DLR strategies can be further optimized to provide innovative and safer therapeutic options for patients with neurological disorders.

Lee et al. found that AuNpRs could function as not only DLR accelerators but also DLR transcriptional factor carriers (Figure 4a) [29]. They hypothesized that DLR transcriptional factors could be loaded inside the pores of AuNpRs, thereby, accelerating the DLR process without administrating viruses. They used several positively charged materials (i.e, cysteamine, PEI, and Mutab) and observed that Mutab was the most stable and nontoxic material with the highest transfection efficiency (Figure 4b,c) [82,83,84]. From these results, they hypothesized that an efficient DLR process could be performed using a virus-free system. This method can avoid host genome integration and virus immunogenicity issues faced in viral transduction systems.

### 2.4. Gene Therapy for SCI Using DLR Technology

#### 2.4.1. SCI: The Incurable Neurological Disorder

Over a decade ago, SCI was considered to lead to a lifetime of medical complications and reliance on a wheelchair, with extremely limited therapeutic options being available [85]. Patients with SCI often received frustrating and hopeless care. Since then, many researchers have been striving to identify the most effective treatment for patients with SCI. However, so far, no therapeutic methodology has been developed. Fortunately, recent advancements in neurosciences have provided new hope [86]. One promising avenue of research is gene therapy using AAV, which offers the potential for regeneration and functional restoration. Although therapeutic methodologies for SCI remain elusive, the prospects for successful treatment continue to improve with each new discovery [57].

In a study, the SCI animal group showed the formation of a physical barrier, known as “glial scar”, at the injury site owing to the recruitment of reactive astrocytes. This scar impeded axonal regeneration and was characterized by the expression of molecules, such as monoamine oxidase B (Figure 5) [87,88,89,90]. Furthermore, certain studies have indicated that premature removal of glial scars can enlarge an injured area and decrease functional recovery in animal models [90,91].

#### 2.4.2. The Potential Therapeutic Method for SCI: AAV Gene Therapy Using DLR Technology

The possibility of using DLR to treat SCI has been studied based on the idea that reactive astrocytes that form glial scars at the injury site may hinder axonal regeneration and lead to neuronal impairment. In vivo DLR targeting of reactive astrocytes to convert them into motor neurons using appropriate transcriptional factors could provide a solution for paralysis in patients with SCI. Yoo et al. successfully identified factors, such as biocompatible materials or specific genetic molecules, which could accelerate the DLR of motor neurons from fibroblasts or astrocytes. Moreover, converting GFAP-positive glial scars to synapsin-positive motor neurons could serve as a solution for the currently incurable SCI (Figure 6).

## 3. Therapeutic Insights and Conclusions

Recently, DLR has withnessed significant advancement. This method enables the direct conversion of differentiated mature cells into various other cell types, without the need for an intermediate pluripotent state. This approach was inspired by the critical role played by transcription factors in the process of converting non-neuronal cells into neurons. For example, the reprogramming of fibroblasts into glutamatergic neurons was first accomplished by overexpressing three transcription factors, namely, ASCL1, BRN2 (also known as Pou3f2), and MYT1L. The expressions of these factors resulted in the formation of neurons capable of expressing neuronal markers and establishing functional synapses that exhibited action potential and spontaneous events [2]. Further studies have been conducted to advance the application of DLR technology to generate human-induced neurons. These studies have shown that the transcription factors ASCL1, BRN2, MYT1L, and neuronal differentiation 1 (NEUROD1) could be used to successfully generate human-induced neurons. The role of NEUROD1 in neuronal development is particularly significant in this process [92]. The ability to directly convert non-neurogenic cells into functional neurons is a significant development in the field of reprogramming. This has been demonstrated by the successful generation of human-induced neurons capable of forming synapses and displaying a functional profile. Moreover, studies have indicated that non-neurogenic cells, such as astroglia, glial cells, and pericytes, could also be converted into functional neurons using this approach [93,94]. Transcription factors have been shown to be effective in converting astroglia into glutamatergic or GABAergic neurons. Additional studies have explored these transcription factors further, leading to the production of other subtypes of neurons, such as dopamine neurons, motor neurons, and intermediate spiny neurons. For example, Caiazzo et al. successfully reprogrammed mouse and human fibroblasts into dopaminergic neurons using ASCL1, NURR1, and LMX1A. These iDNs were functionally similar to natural dopaminergic neurons [95]. Furthermore, recent findings have suggested that the use of miRNA could be a promising approach for direct reprogramming [19,20]. Similar to transcription factors, a combination of miRNAs can be used to reprogram fibroblasts into functional neurons [19]. Additionally, the presence of transcription factors can enhance the role of miRNAs in neural reprogramming. For instance, studies have demonstrated that miR-9/9*, miR-124, or miR-218 in combination with other transcription factors, such as ISL1 and LHX3, could promote the reprogramming of fibroblasts into motor neurons [19,96]. MirR-34b/c cotreated with ASCL1 and NURR1 enhanced the generation of iDNs by activating the Wnt1 signaling cascade [97]. A mixture of miR-218 and other transcription factors, such as ASCL1, NEUROD1, and LMX1A, promoted the conversion of astrocytes to dopaminergic neurons in vitro and in vivo [13]. Collectively, these findings highlight the crucial role of both transcription factors and miRNAs in directing the fate of neuronal cells.

A recent study has suggested that the combination of miRNAs with transcription factors contributes to regulating ROS in vivo and in vitro. miRNA-containing nanoparticles are known to reduce ROS. According to a previous study, polyketal (PK3) nanoparticles that contained a mixture of miR-106B, miR-148b, and miR-204 significantly improved cardiac function after myocardial infarction by reducing ROS, which was enhanced by nicotinamide adenine dinucleotide phosphate oxidase [98]. In DLR research, AuNpRs containing the viral transcription factors, ASCL1, NURR1, PITX3, and LMX1A have significantly increased TH-positive neurons by scavenging ROS molecules, which led to the conversion of dopaminergic neurons in PD mouse model. Given the possibility that nanoparticles containing miRNA and transcription factors can reduce ROS in vivo, nanoparticle-based ROS removal may play a vital role in DLR.

The human brain has limited regenerative capacity, indicating that it is unable to naturally replace damaged neurons at the same rate as other organs. This limitation could be attributed to the fact that the process of neurogenesis, or the growth and development of new neurons, is more complex and tightly regulated in the brain than in other tissues of the body [99]. Hence, alternative cell sources are required for treating central nervous system diseases. In this regard, in vivo reprogramming has emerged as a promising therapeutic approach. However, for successful reprogramming, ensuring precise targeting of specific cell populations is crucial. AAV-mediated gene therapy is a viable method for regulating the final cell type during in vivo reprogramming, which could lead to optimal therapeutic outcomes for central nervous system diseases.

Meanwhile, the use of transcription factors in DLR has faced limitations in clinical translation, due to the fact that obtaining sufficient cells is difficult owing to low efficiency and lack of maturity. Therefore, the development of new strategies, such as the use of ROS scavenging molecules and biocompatible nanomaterials, such as AuNpRs, have opened new avenues in the field of gene therapy for incurable diseases. Further research in this area could lead to breakthrough treatments that would improve the quality of life of patients with central nervous system diseases.

## Figures and Tables

**Figure 1 nanomaterials-13-01680-f001:**
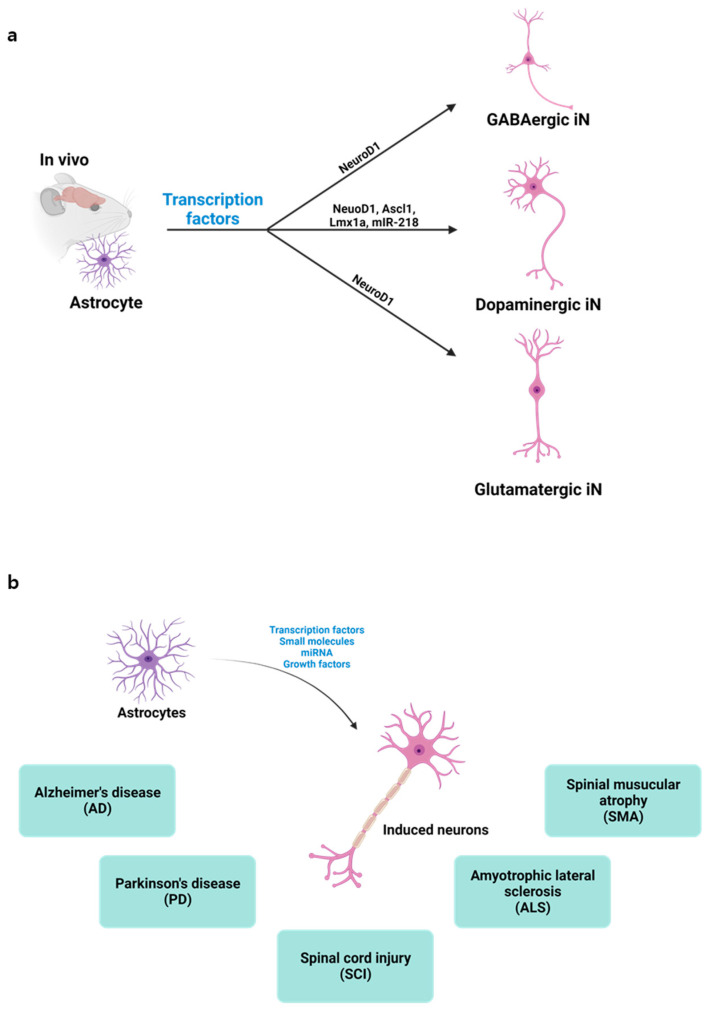
Direct conversion of astrocytes into induced neurons: (**a**) Depiction of in vivo astrocyte reprogramming into various neuron types, i.e., GABAergic, dopaminergic, or glutamatergic, achieved through the targeted expression of specific transcription factors; (**b**) Illustration showcasing the potential applications of in vivo direct neuronal reprogramming.

**Figure 2 nanomaterials-13-01680-f002:**
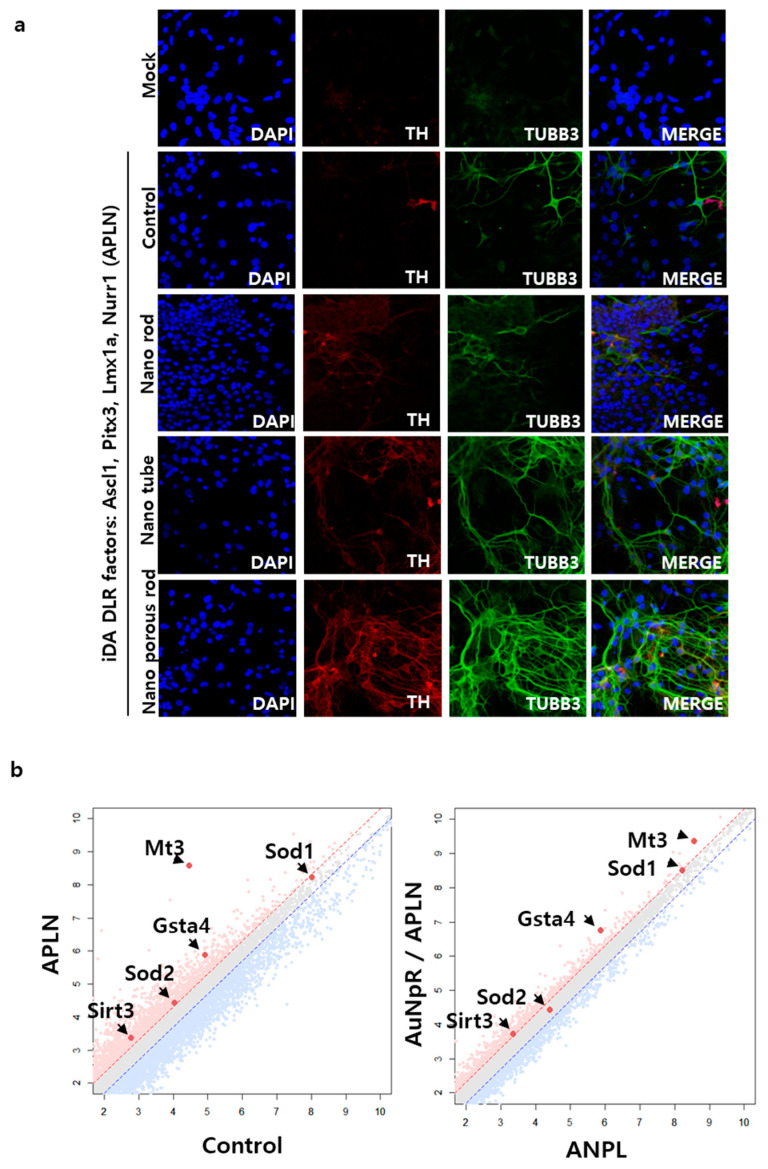
Efficient iDA neuron direct lineage reprogramming using AuNpRs: (**a**) Immunostaining analysis of various types of nanomaterials (gold nanorod, gold nanotube, gold nanoporous-rod) induced direct reprogrammed iDA neurons, immunofluorescent staining of induced neuron for neural markers, TH and TUBB3 (Scale bar = 40 µm); Data from Lee et al. 2022, Acta Biomaterialia [29]. Copyright from {2022}Acta Biomaterialia. (**b**) the dot plots representing the relative expression levels of control fibroblasts vs. APLN transduced fibroblasts (left) and APLN transduced fibroblast vs. APLN transduced fibroblast with AuNpR treating. Red dots, upregulated gene and blue dots, downregulated gene; Data from Lee et al. 2022, Acta Biomaterialia [29]. Copyright from {2022}Acta Biomaterialia.

**Figure 3 nanomaterials-13-01680-f003:**
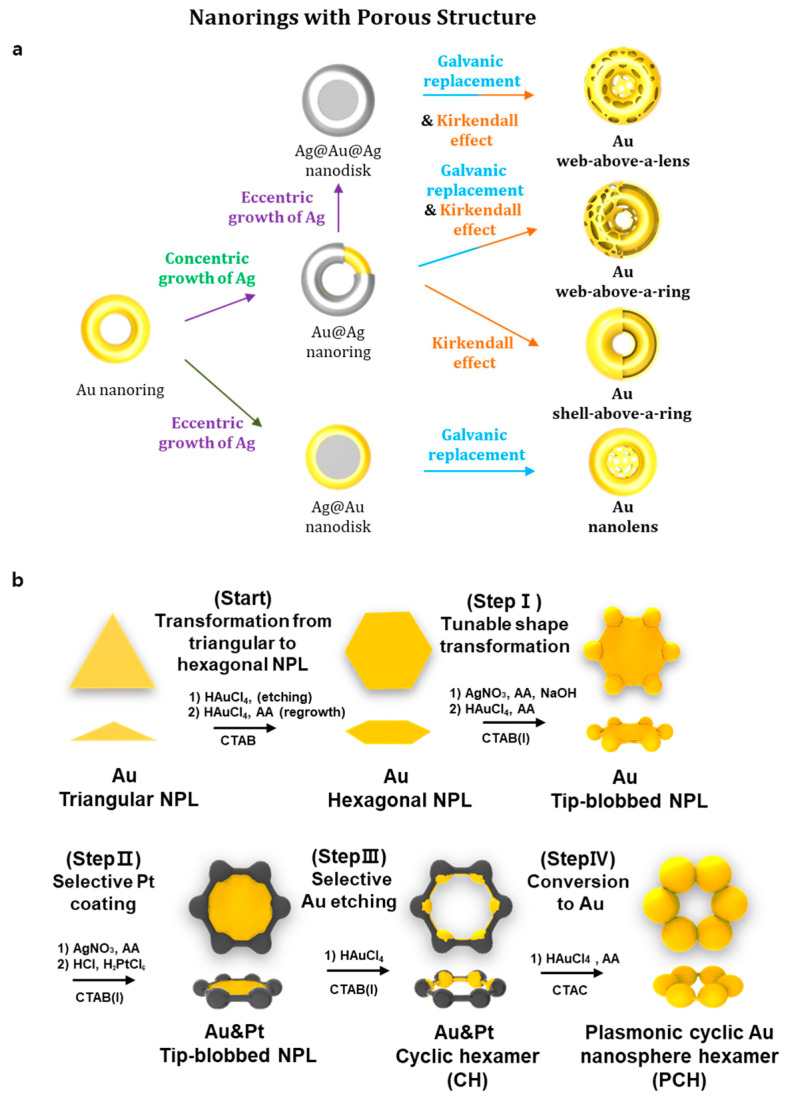
Electro mechanism-based nanoparticle fabrication method: (**a**) The method of fabricating porous structure integrated gold nanorings; Data from Lee et al. 2022, Acta Biomaterialia [29]. Copyright from {2022}Acta Biomaterialia.; (**b**) the method to fabricate nanosphere hexamer; Data from Lee et al. 2022, Acta Biomaterialia [29]. Copyright from {2022}Acta Biomaterialia.

**Figure 4 nanomaterials-13-01680-f004:**
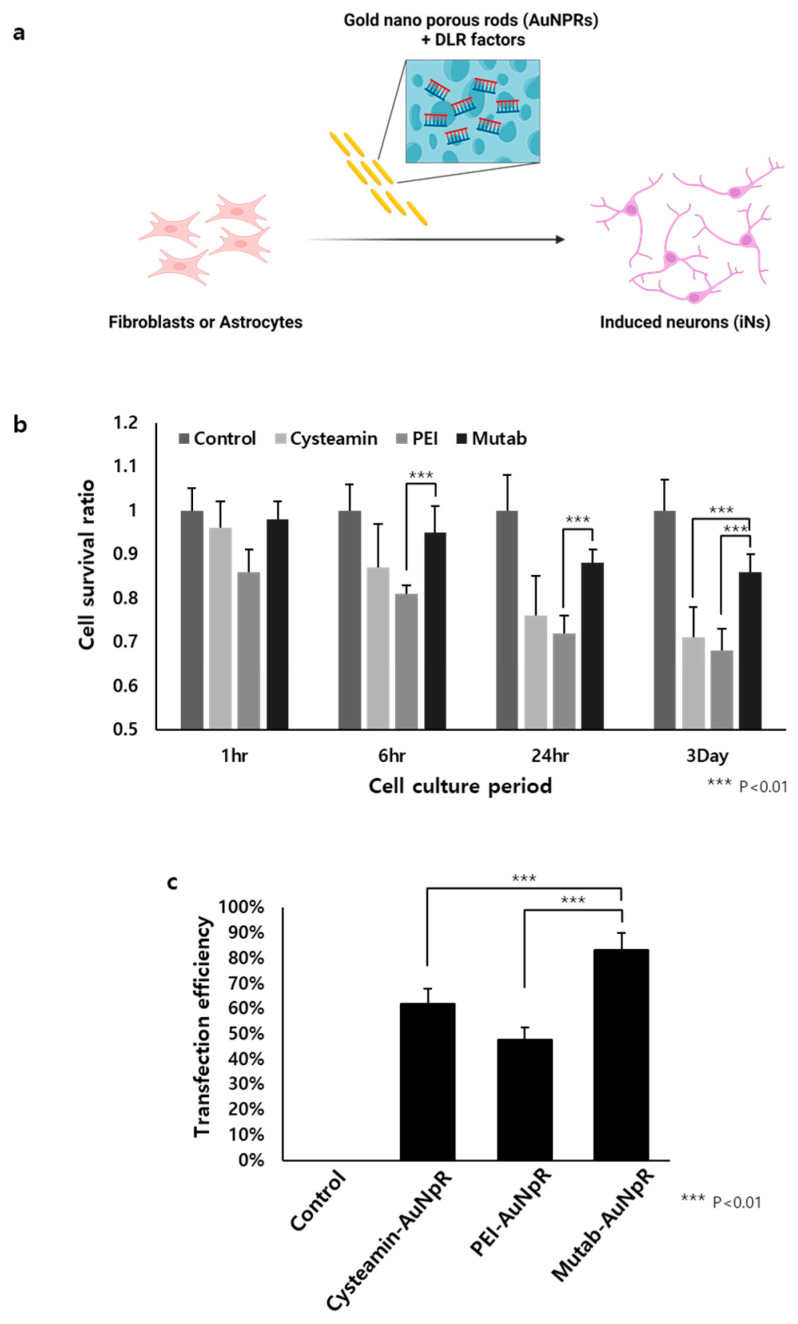
Non-viral nanoparticle-based gene delivery system: (**a**) The schematic illustration of non-viral nanoparticle-based iNs DLR; (**b**) cell survival ratio in accordance with using various type of positive charged nanoparticle coating materials (cysteamine, PEI, and Mutab); (**c**) transfection efficiency for using different type of nanoparticle coating materials.

**Figure 5 nanomaterials-13-01680-f005:**
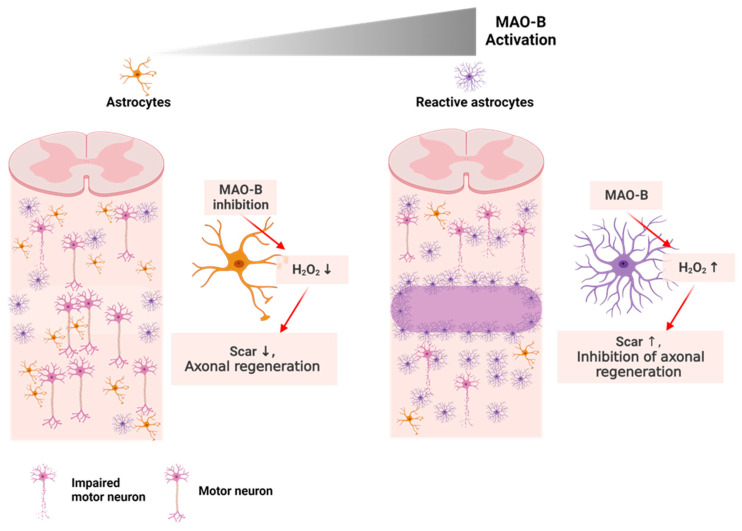
Schematic illustration showing the consequence of spinal cord injury in neuron and astrocyte.

**Figure 6 nanomaterials-13-01680-f006:**
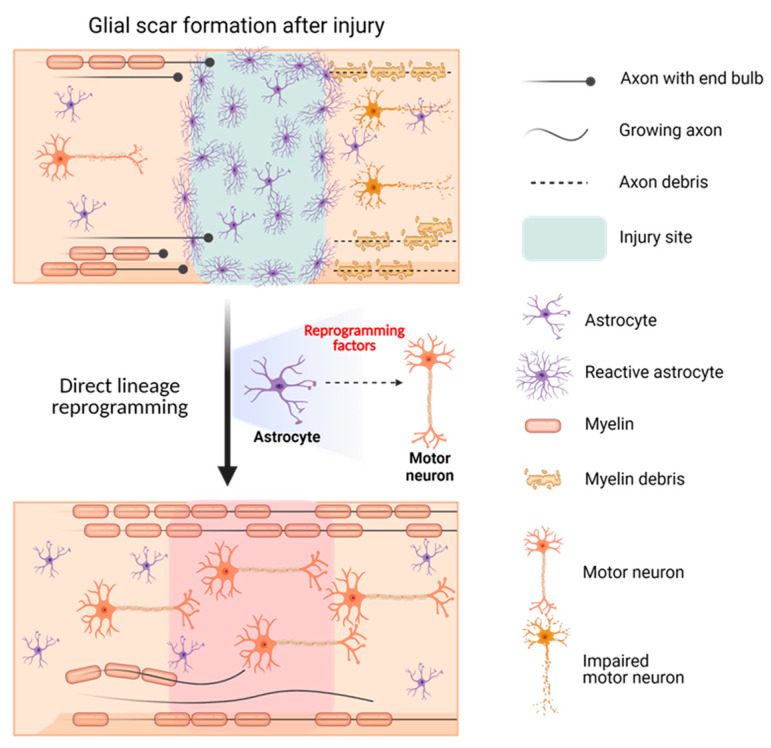
Schematic illustration showing the effect of motor neuron DLR on spinal cord injury.

**Table 1 nanomaterials-13-01680-t001:** Table showing the various type of DLR accelerators.

Species	Starting Cells	Target Cells	Materials	Efficiency	Ref.
Ms/Hu	Fibroblasts	Neurons	Small molecules (CHIR, LDN, AA)	~35%	[22]
Ms/Hu	Fibroblasts	Neurons	miRNA	~40%	[4]
Ms/hu	Fibroblasts	Dopaminergic neurons	Electromagnetized gold nanoparticles	~55%	[5]
Ms	Fibroblasts	Dopaminergic neurons	Electromagnetized graphene nanosheet	~20%	[31]
Ms/Hu	Fibroblasts	Dopaminergic neurons	Elongated nanoporous gold nanorod	~40%	[29]
Hu	Fibroblasts	Neurons	Polymer-functionalized Nanodot	~40%	[46]

**Table 2 nanomaterials-13-01680-t002:** Table showing the AAV2-based ongoing and completed clinical trials.

No.	Name	Application	Phase	Status	Identifier	Ref.
1	AAV-SMN1	Muscular Atrophy, Spinal	Phase 4	Active, not recruiting,	NCT05073133	[60]
2	AAV2-BDNF	Alzheimer’s disease	Phase 1	Recruiting	NCT05040217	[61]
3	AAV2-GDNF	Parkinson’s disease	Phase 1	Recruiting	NCT04167540	[62]
4	AAV2-hRPE65v2	Inherited retinal dystrophy	-	Active, not recruiting	NCT03602820	[58,63]
5	AAV2/5-RPGR	X-linked retinitis pigmentosa	Phase 1/2	Completed	NCT03252847	[69]
6	rAAV2.REP1	Choroideremia	Phase 2	Completed	NCT02671539	[58]
7	AAV2-REP1	Choroideremia	Phase 2	Completed	NCT02553135	[64]
8	AAV2-hAQP1	Squamous cell head and neck cancer Radiation induced xerostomia Salivary hypofunction	Phase 1	Recruiting	NCT02446249	[65]
9	AAV2-hCHM	Choroideremia	Phase 1/2	Active, not recruiting	NCT02341807	[66]
10	rAAV2.REP1	Choroideremia	Phase 1/2	Completed	NCT02077361	[58]
11	AAV2-GDNF	Parkinson’s disease	Phase 1	Completed	NCT01621581	[62]
12	AAV2-sFLT01	Macular degeneration	Phase 1	Completed	NCT01024998	[67]
13	AAV2-NTN	Parkinson’s disease	Phase 1	Completed	NCT00252850	[68]

## Data Availability

Data are available upon request and can be accessed by contacting the corresponding author.

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
