# Peer review of "Gene Therapy Using Efficient Direct Lineage Reprogramming Technology for Neurological Diseases"

_nanomaterials, 2023, doi:10.3390/nano13101680_

Round 1
Reviewer 1 Report (Previous Reviewer 2)
The topic of this article is interesting, but not a newness, the authors presenting some possibilities of use the efficient direct lineage reprogramming technology for neurological diseases gene therapy.
While the rationale of this study is strongly supported by past literatures, and seems to possess great potential, the manuscript at its current state holds critical concerns regarding its novelty and significance of information provided.
After reading the manuscript, the following doubts and suggestions have arisen:
In the introduction, the authors should highlight the experimental and clinical importance of different nanotechnology methods implementation in the biomedical engineering field.
This review article should be more comprehensive, providing supplementary background regarding the current literature data about the various possibilities and application of gene manipulation in the treatment of different neurodegenerative diseases, revealing the benefits of the optimization the direct lineage reprogramming technology. Literature analysis shows various communicated data in the field, which should be cited, for example:
· Vasan L et al. Direct neuronal reprogramming: bridging the gap between basic science and clinical application. Front Cell Dev Biol 2021; 9: 681087
· Bajohr J, Faiz M. Direct Lineage Reprogramming in the CNS. Adv Exp Med Biol 2020; 1212: 31-48.
· Zhang Y et al. Prospects of directly reprogrammed adult human neurons for neurodegenerative disease modeling and drug discovery: iN vs. iPSCs Models. Front Neurosci 2020; 14: 546484.
· Vignoles R et al. Direct lineage reprogramming for brain repair: breakthroughs and challenges. Trends Mol Med 2019, 25 (10): 897- 914.
and others.
Some other aspects were found in this manuscript:
- some of the abbreviations are not explained, which creates a lot of confusion and makes the text difficult to read and understand;
- the authors should mandatory upgrade the references;
- the name of the journals should be abbreviated according to the Index Medicus (i.e. 4, 5, 10, 11 and many others)
- spelling check of the text is mandatory;
- English including grammar, style and syntax, should be improved through the professional help from English Editing Company for Scientific Writings.
English including grammar, style and syntax, should be improved through the professional help from English Editing Company for Scientific Writings.
Author Response
Dear reviewer,
First of all we would like to say thank you for giving us the valuable comments. As reviewer suggested, we are resubmitting the manuscript with making well-correction.
We have carefully addressed each of their suggestions and made significant improvements to our manuscript accordingly. To ensure the quality and professionalism of the language, we enlisted the services of a professional English editing company, Enago. The editing certificate has been attached with revised manuscript.
We are confident that the revisions have enhanced the manuscript and made it suitable for publication in Nanomaterials. We appreciate your consideration and look forward to your response.
Additionally, we have attached the questions by questions which reviewers has raised in the below.
<Attached files are including Main figures, revised manuscript (revision tracked), Publication licenses for figure 1 and English editing certificates.>
Best regard,
Junsang Yoo, Ph.D.
- In the introduction, the authors should highlight the experimental and clinical importance of different nanotechnology methods implementation in the biomedical engineering field.
→ Thank you for your valuable feedback. We have included content of importance of nanotechnology in introduction section in the line 150-163.
- This review article should be more comprehensive, providing supplementary background regarding the current literature data about the various possibilities and application of gene manipulation in the treatment of different neurodegenerative diseases, revealing the benefits of the optimization the direct lineage reprogramming technology. Literature analysis shows various communicated data in the field, which should be cited, for example:
→ As reviewer suggested, we have added additional explanation, references, and figures.
- some of the abbreviations are not explained, which creates a lot of confusion and makes the text difficult to read and understand;
→ We apologize for the confusion. We have revised the abbreviations in the revised manuscript.
- the name of the journals should be abbreviated according to the Index Medicus (i.e. 4, 5, 10, 11 and many others)
→ We have checked and revised the references format throughout the revised manuscript.
- spelling check of the text is mandatory
→ Thank you for your comment and sorry for the confusion. We have re-checked the spelling and corrected the miss-spelled words.
- English including grammar, style and syntax, should be improved through the professional help from English Editing Company for Scientific Writings.
→ Thank you for your valuable comment. We would like to assure the reviewer that we have taken their concerns regarding English language issues in the manuscript seriously. As a result, we have engaged the professional English editing company, Enago, to thoroughly review and revise our manuscript twice. After these two rounds of editing, we are confident that all language issues have been addressed and the manuscript is now clear and well-polished. Also we have attached the editing certificate for that.

Reviewer 2 Report (Previous Reviewer 3)
This well-organised and informative review can be an excellent addition to the scientific community. I recommend it for publication.
Author Response
Dear reviewer,
Thank you very much for your nice comment.
Best regard,
Junsang Yoo, Ph.D.
Reviewer 3 Report (Previous Reviewer 1)
The authors addressed the previous review comments in the response letter and significantly improved the manuscript. Therefore, I recommend the manuscript for publication after minor revision.
Lines 103-111. The authors highlighted biocompatible materials as novel diagnostic tools. However, the entire manuscript was focused on gene manipulation for direct lineage reprogramming. Please consider focusing only gene/drug delivery to improve clarity.
Line 121. Please clarify “small-molecule-based delivery system”. Does this mean small molecule drug delivery system?
Lines 141-143. Please clarify this sentence. Can reducing oxidative stress increase the efficiency of the DLR process?
Lines 207-211. The authors highlighted AAV2 as a safe and efficient in vivo gene therapy approach. However, AAV-based gene delivery still has concerns about their potential immunogenicity and genome integration concerns. Please consider briefly mentioning the current concerns about AAV vectors, and it would support efforts for non-viral vector development.
Lines 222-223. Does this sentence mean AuNpRs can induce the expression of the 4 genes in vivo? Please improve clarity.
Lines 363-364. Do the AuNpRs carry proteins of the 4 transcription factors or mRNA?
Author Response
Dear reviewer,
Thank you very much for your nice comment and nice point out. We have carefully addressed each of your suggestions and made significant improvements to our manuscript.
We have attached the questions by questions which reviewer has raised in the below.
Best regard,
Junsang Yoo, Ph.D.
Lines 103-111. The authors highlighted biocompatible materials as novel diagnostic tools. However, the entire manuscript was focused on gene manipulation for direct lineage reprogramming. Please consider focusing only gene/drug delivery to improve clarity.
→ We removed this paragraph from our manuscript to provide clarity on gene manipulation for direct lineage reprogramming.
Line 121. Please clarify “small-molecule-based delivery system”. Does this mean small molecule drug delivery system?
→ We have revised the manuscript more specifically.
Lines 141-143. Please clarify this sentence. Can reducing oxidative stress increase the efficiency of the DLR process?
→ We have revised the sentence.
Lines 207-211. The authors highlighted AAV2 as a safe and efficient in vivo gene therapy approach. However, AAV-based gene delivery still has concerns about their potential immunogenicity and genome integration concerns. Please consider briefly mentioning the current concerns about AAV vectors, and it would support efforts for non-viral vector development.
→ We have included this information of the revised manuscript.
Lines 222-223. Does this sentence mean AuNpRs can induce the expression of the 4 genes in vivo? Please improve clarity.
→ We have included this information of the revised manuscript.
Lines 363-364. Do the AuNpRs carry proteins of the 4 transcription factors or mRNA?
→ We have revised the manuscript more specifically. AuNpRs carry viral transcription factor, ASCL1, NURR1, PITX3, and LMX1A.

Round 2
Reviewer 1 Report (Previous Reviewer 2)
The authors have significantly revised the manuscript addressing the concern raised. I consider it could be accepted for publication in this journal.
Author Response
Dear reviewer,
We are deeply appreciative of your positive evaluation of our revised manuscript. Your insightful feedback played a significant role in the enhancements made to the manuscript, and we are sincerely grateful for the time and effort you invested in this process. Your approval of the improvements is a testament to the value of your contributions.
Best regards,
J Yoo
This manuscript is a resubmission of an earlier submission. The following is a list of the peer review reports and author responses from that submission.
Round 1
Reviewer 1 Report
Chang and Lee et al. reviewed the proposed technologies, nanoporous Au nanoparticle-based gene delivery systems, to enhance the efficiency of the direct lineage reprogramming (DLR) process. The contents are novel and potentially significant in the clinical treatment of spinal cord injury (SCI). However, the manuscript needs to improve its clarity. The detailed comments are listed below.
Line 49. Please briefly explain why the DLR could be more cost-effective than iPSC re-differentiation.
Lines 89-91. Please explain more possible mechanisms of these studies. For example, how can these nanomaterials improve the efficiency of DLR?
Line 102-104. Please rewrite this sentence to improve clarity.
Figure 1. Please add more information in the figure legend so readers can easily understand the data. For example, why were TH and TUBB3 stained in (a)? What do blue and red dots indicate in (b)? APLN transduced fibroblasts?
Figure 2. Please add more information in the figure legend. Also, please add the figure legend of Supplementary Figure 1.
Lines 152, 155-156. The subtitle shows DNA, but please specify the delivery cargo for DLR. Figure 3a seems that double-strand nucleic acids are included in AuNPRs. Are these DNA, mRNA, or miRNA?
Lines 216-218. The authors described that the combination of miRNAs had shown the ability to reprogram fibroblasts into functional neurons. Does this indicate the expression of transcription factors and specific miRNAs independently contributed to DLR? The connection between transcription factors, miRNAs, and nanomaterial-based ROS removal is unclear to this reviewer. Please improve clarity on this point.
Line2 234-237. Please clarify what the limitations of DLR approaches in the clinical translation are.
Figure 4 and 5. Please add more information in the figure legends to improve clarity.
Reviewer 2 Report
Yujung Chang and co. aimed to reveal the different possibilities of use the efficient direct lineage reprogramming technology for neurological diseases gene therapy.
However, I believe that this manuscript (in the present form) does not meet the criteria to be published in this journal, as I have identified some important flaws:
The abstract should be improved. It can be more concise and clearly state the main concept of the review.
The introduction section should be more complete, providing supplementary background in the field. At the end of introduction, it would be better to add a few sentences to explain the main focus of the review.
This review paper should be more complete, providing supplementary background regarding the recent communicated data about the numerous possibilities and application of nanotechnology for neurological diseases gene therapy, as well as the advantages of accelerate the reprogramming efficiency in the optimization of the therapeutic use. Literature analysis reveals various communicated data in the field should be cited.
Some other aspects were found in this manuscript:
- all abbreviations should be expanded in the first appearance and should not be repeated, in order to decongest the text and facilitate the understanding of the information transmitted;
- some of the abbreviations are not explained, which creates a lot of confusion and makes the text difficult to read and understand;
- the authors should mandatory upgrade the references;
- the name of the journals should be abbreviated according to the Index Medicus;
- at the references the authors should provide the DOI of the articles;
- spelling check of the text is required;
- overall revision regarding grammatical errors, style and syntax and general use of English is recommended.
Reviewer 3 Report
In this review article, the authors intended to overview current progress in gene therapy by applying direct lineage reprogramming technology. While this research direction is a promising and attractive approach to accelerate the progress in the field, the authors still need to open the whole potential of this topic in the paper. Even though the illustration materials provided by the authors are informative and well-executed, the text appears chaotic and unstructured. For example, the basic explanation of the whole concept of direct lineage reprogramming technology first appears in full only in the section ‘therapeutic insights and conclusions’ instead of straight in the introduction section. There are numerous places in the text where sentences are not linked to each other, many odd uses of English words, and grammar errors. The examples for applicability of the described approaches seemed random for some brain pathologies or spinal cord injuries instead of making a clear logic for a vast implementation of the technique in various neurological diseases across the central nervous system. The authors must put more effort into working on the text to keep it meaningful, logical and useful not only for researchers whose work in the field but ideally for a much wider auditorium of readers.